# CAMMARL: CONFORMAL ACTION MODELING IN MULTI AGENT REINFORCEMENT LEARNING

## ABSTRACT

Before taking actions in an environment with more than one intelligent agent, an autonomous agent may benefit from reasoning about the other agents and utilizing a notion of a guarantee or confidence about the behavior of the system. In this article, we propose a novel multi-agent reinforcement learning (MARL) algorithm CAMMARL, which involves modeling the actions of other agents in different situations in the form of confident sets, i.e., sets containing their true actions with a high probability. We then use these estimates to inform an agent's decision-making. For estimating such sets, we use the concept of conformal predictions, by means of which, we not only obtain an estimate of the most probable outcome but get to quantify the operable uncertainty as well. For instance, we can predict a set that provably covers the true predictions with high probabilities (e.g., 95%). Through several experiments in two fully cooperative multi-agent tasks, we show that CAMMARL elevates the capabilities of an autonomous agent in MARL by modeling conformal prediction sets over the behavior of other agents in the environment and utilizing such estimates to enhance its policy learning.

## 1 INTRODUCTION

Developing systems of autonomous agents capable of effective multi-agent interactions can be very useful in modern cooperative artificial intelligence (AI). For instance, service robots, surveillance agents, and many more similar applications require profound collaboration among agents (and with humans), without prior coordination. Now, to enable complex, constructive behaviors to emerge from unsupervised interactions among agents, an essential skill for an agent to have is the ability to reason about other agents in the environment. There has been considerable research addressing this problem of *agent* or *opponent modeling* (Albrecht & Stone, 2018). Generally, it involves constructing models of other agents that learn useful attributes to inform its own decision-making (such as the future actions of the other agents, or their current goals and plans) from current or past interaction history (such as the previous actions taken by other agents in different situations).

We are interested in the particular aspect of an interactive, autonomous agent that involves learning an additional, independent model to make predictions about the actions of the other agents in the environment, supplemental to its reinforcement learning-based policy to make decisions related to its downstream task. An autonomous agent can then incorporate those estimates to inform its decision-making and optimize its interaction with the other agents. While there exist several methods for developing such models for other agents (Albrecht & Stone, 2018), there is currently no method or theory to the best of our knowledge that would allow an agent to consider the correctness or confidence of the predictions of the learned model.

**Conformal Predictions.** Conformal predictions or inference is a fitting method for generating statistically accurate uncertainty sets for the predictions from machine learning classifiers. It is steadily gaining popularity owing to its explicit and non-asymptotic guarantees over the produced sets (Angelopoulos & Bates, 2021). In other words, we can obtain conformal sets that provably contain the true predictions with high probabilities, such as 95%, chosen in advance. This can be very useful and successful in high-risk learning settings, especially in decision-making in medical applications from diagnostic information, for instance, which demand quantifying uncertainties to avoid insufferable model failures. What if we only prefer to use the predictions when the model is confident? For example, doctors may only consider a predicted medical diagnosis when the model

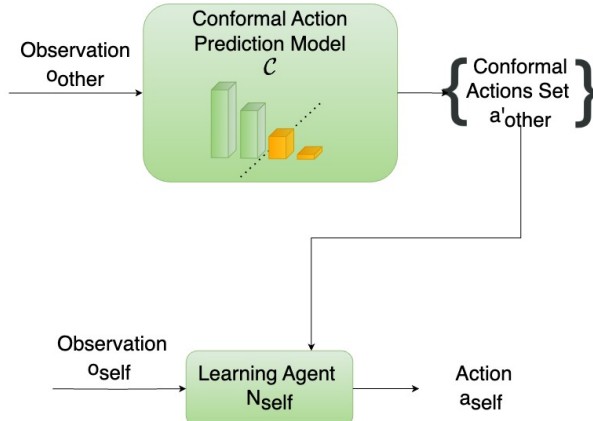

Figure 1: Our proposed methodology of informing an autonomous agent's decision-making by means of conformal predictions of action sets of other agents in the environment illustrated with two agents for simplicity. Two agents ($\mathcal{N}_{self}$, $\mathcal{N}_{other}$) receive their own partial observations from the environment ($o_{self}$, $o_{other}$) and take their actions ($a_{self}$, $a_{other}$). An independent conformal action prediction model $\mathcal{C}$ learns to output a conformal action set, $\{a'_{other}\}$, corresponding to $\mathcal{N}_{other}$ which are then used as additional inputs for training by $\mathcal{N}_{self}$ to inform its policy and perform its action $a_{self}$.

is at least 95% accurate, or may want to use the predicted set with high credence to consider ruling out relevant possibilities. So, in this article, we aim to enhance the capabilities of an agent in a multi-agent reinforcement learning (MARL) setting by modeling and using conformal prediction sets (or the latent representations learned in the process[1]) over the behavior of an autonomous system. In particular, we model other agents' actions in the form of confident sets, i.e., sets that contain other agents' true actions with a high probability. We hypothesize that these estimated conformal sets would inform our *learning* agent's decision-making and elevate its performance in MARL. Figure 1 shows the high-level idea of our proposed model for learning agents in any given environment.

In this work, we aim to introduce a novel framework to train an autonomous agent that enhances its decision-making by modeling and predicting *confident conformal* actions of other agents in the environment — the CAMMARL algorithm (Section 3), and then empirically demonstrate that conformal action modeling used in CAMMARL indeed can help make significant improvements in cooperative policies learned by reinforcement learning agents in two multi-agent domains (Section 4).

## 2 RELATED WORKS

Decision-making without reasoning about other agents in the environment can be very challenging, for instance, due to weak or no theoretical guarantees, non-stationarity (single agent's perspective), and inefficient coordination for a considerable coherent joint behavior (Matignon et al., 2012). Modeling other agents in an environment is not new and has been studied in the past (Albrecht & Stone, 2018; Albrecht et al., 2020). However, our proposal of predicting conformal sets of actions of the other agents in the environment (with high probability) is novel and has not been attempted to the best of our knowledge.

**Learning world models.** Model-based reinforcement learning (MBRL) has certainly shown its advantages in data efficiency, generalization, exploration, counterfactual reasoning, and performance in many tasks and domains (Hafner et al., 2020; 2021; Jain et al., 2022; Moerland et al., 2023; Pal & Leon, 2020; Polydoros & Nalpantidis, 2017) in single-agent RL, and now, it has also started to attract attention in MARL (Wang et al., 2022). However, most of the current works in model-based MARL do not yet focus on teammate or opponent modeling. Some recent works (Park et al., 2019b; Zhang et al., 2021) incorporated dynamics modeling and a prediction module to estimate the actions

---

[1]More details in Appendix C.

of other agents within the construction of the environment model. However, these prediction models were trained without accessing the true trajectories from the other agents which can be problematic in several use cases.

**Learning agent models.** A widely popular technique to reason about other agents in the environment is to learn representations of different properties of other agents. For instance, learning to reconstruct the actions of other agents from their partial observations (He et al., 2016; Mealing & Shapiro, 2015; Panella & Gmytrasiewicz, 2017; Albrecht & Ramamoorthy, 2015), modeling an agent or its policy using encoder-decoder-based architectures Grover et al. (2018); Zintgraf et al. (2021), learning latent representations from local information with or without utilizing the modeled agent's trajectories (Papoudakis et al., 2021; Xie et al., 2021) or modeling the forward dynamics of the system through relational reasoning using graph neural networks (Tacchetti et al., 2018). *Theory-of-Mind Network* or TomNet learned embeddings corresponding to other agents in the environment for meta-learning (Rabinowitz et al., 2018). Some works also constructed I-POMDPs to utilize recursive reasoning (Albrecht & Stone, 2018) assuming unrestricted knowledge of the observation models of other agents. Nevertheless, CAMMARL involves no form of reconstruction of other agent's policy or rewards, or state models. Any of these techniques can be used with CAMMARL which, however, is not the objective of this work. Also, unlike CAMMARL, many of these aforementioned techniques evaluate in fully-observable environments or rely upon direct access to other agents' experience trajectories even during execution. This can be infeasible in various settings.

**Multi-agent reinforcement learning (MARL).** Numerous deep MARL research works that focus on partial observability in fully cooperative settings indirectly involve reasoning about the intentions of teammates or opponents in an environment (Gronauer & Diepold, 2022). For instance, many works allow agents to communicate, enabling them to indirectly reason about the others' intentions (Lazaridou et al., 2016; Foerster et al., 2016; Sukhbaatar et al., 2016; Das et al., 2017; Mordatch & Abbeel, 2018; Gupta et al., 2021; Zhu et al., 2022). On the other hand, some studied the emergence of cooperative and competitive behaviors among agents in varying environmental factors, for instance, task types or reward structures (Leibo et al., 2017). Recent work in hierarchical reinforcement learning also attempts to develop a hierarchical model to enable agents to strategically decide whether to cooperate or compete with others in the environment and then execute respective planning programs (Kleiman-Weiner et al., 2016). However, none of these works study the improvement in an autonomous agent's decision-making via directly modeling the other agents in the environment or predicting their actions or current or future intentions.

**Inverse reinforcement learning (IRL).** Research in the field of IRL also relates to our work because we share the key motive of inferring other agents' intentions and then use it to learn a policy that maximizes the utility of our learning agent (Arora & Doshi, 2021). However, IRL addresses this by deducing the reward functions of other agents based on their behavior, assuming it to be nearly optimal. On the other hand, in CAMMARL we directly model the other agent's actions based on their observations and use these estimates to indirectly infer their goal in an online manner.

**Conformal prediction.** Estimating well-grounded uncertainty in predictions is a difficult and unsolved problem and there have been numerous approaches for approximating it in research in supervised learning (Gawlikowski et al., 2021). Recent works in conformal predictions (Angelopoulos et al., 2020; Lei et al., 2018; Hechtlinger et al., 2018; Park et al., 2019a; Cauchois et al., 2020; Messoudi et al., 2020) have now significantly improved upon some of the early research (Vovk et al., 2005; Platt et al., 1999; Papadopoulos et al., 2002), for instance in terms of predicted set sizes, improved efficiency, and providing formal guarantees. For this article, we adapt the core ideas from *Regularized Adaptive Prediction Sets (RAPS)* (Angelopoulos et al., 2020) to our setting owing to its demonstrated improved performance evaluation on classification benchmarks in supervised learning (Angelopoulos et al., 2020). Key description of conformal prediction is in Appendix B.

## 3 THE CAMMARL ALGORITHM

### 3.1 MOTIVATION

The primary motivation behind CAMMARL is to enhance an agent's decision-making in multi-agent environments by incorporating predictions about other agents' actions. CAMMARL utilizes conformal prediction, a technique that provides statistically valid uncertainty quantification. In the

context of CAMMARL, conformal prediction allows us to generate sets of possible actions for other agents, with a guaranteed probability of containing the true action. This approach offers a principled way to represent uncertainty in action predictions since the prediction sets are automatically calibrated, ensuring that the specified coverage probability is achieved in practice.

## 3.2 MATHEMATICAL MODEL

Formally, we consider two agents in the environment — learning agent denoted by *self* and the other agent denoted by *other*. The partially observable Markov game Littman (1994) for our setting can then be defined using the following tuple[2]:

$$\langle \mathcal{N}_i, \mathcal{S}, \mathcal{A}_i, \mathcal{O}_i, \mathcal{T}, \mathcal{C}, \pi_{\theta_i}, r_i \rangle_{i \in \{self, other_1 \ldots other_{K-1}\}}$$

With the set $\mathcal{S}$ describing the possible true states (or full observations) of the environment, $K$ agents, $\mathcal{N}_{self}$ and $(K-1)$ $\mathcal{N}_{other_j}$ ($j \in [1, K]$), observe the environment locally using their sets of observations $\mathcal{O}_{self}$ and $(K-1)$ $\mathcal{O}_{other_j}$ respectively, and act using their set of actions, $\mathcal{A}_{self}$ and $(K-1)$ $\mathcal{A}_{other_j}$. Each agent $i$ can select an action $a_i \in \mathcal{A}_i$ using their policy $\pi_{\theta_i}$, and their joint action $\mathbf{a} \in \mathcal{A}_{self} \times \mathcal{A}_{other_1} \times \ldots \mathcal{A}_{other_K}$ then imposes a transition to the next state in the environment according to the state transition function $\mathcal{T}$, defined as a probability distribution on the subsequent state based on current state and actions, $\mathcal{T}: \mathcal{S} \times \mathcal{A}_{self} \times \mathcal{A}_{other_1} \times \ldots \mathcal{A}_{other_K} \times \mathcal{S} \rightarrow [0, 1]$. The agents use their individual reward function $r_i(s, a) : \mathcal{O}_i \times \mathcal{A}_i \rightarrow \mathbb{R}$. All agents aim to maximize their own total expected rewards $R_i = \sum_{t=0}^{T} \gamma^t r_i^t$ where $\gamma \in [0, 1)$ as the discount factor and T is the time horizon.

In CAMMARL, at each time step *t*, we also use a conformal prediction model for the $j$-th agent, defined as a set-valued function, $\mathcal{C} : \mathbb{R}^d \rightarrow 2^{|\mathcal{A}_{other_j}|}$

$$\mathcal{C}(o_{other_j}^t) \rightarrow \{A_{other_j}^i\}$$

which outputs a conformal action predictive set $\{A_{other_j}^t\}$ for each input of $\mathcal{N}_{other_j}$'s local observation $o_{other_j}^t \in \mathcal{O}_{other_j}$ at the time.

## 3.3 CONFORMAL ACTION MODELING

---
**Algorithm 1** CONFORMAL ACTION MODELING IN MARL
---
1: $N_{self}, N_{other_j} \leftarrow$ Initialize Actor-Critic networks for — $\mathcal{N}_{self}$ and $\mathcal{N}_{other_j}$, where $j \in [1, K]$
2: (K-1) `conformalModels` $\leftarrow$ Initialize the conformal model to predict conformal action sets
3: **for** `episode` $= 1, 2, \ldots$ **do**
4:     Fetch observations $o_{self}, o_{other_1} \ldots o_{other_K}$ from environment
5:     **for** `timesteps` $= 1, 2, \ldots, T$ **do**
6:         `conformalActions` $\leftarrow$ `conformalModels`$(o_{other_j})$; for $j \in [1, K]$    ▷ Predict conformal action set
7:         $o_{self} \leftarrow o_{self} +$ `conformalActions`    ▷ Concatenate conformal actions to $o_{self}$
8:         Run agent policies in the environment
9:         Collect trajectories of $\mathcal{N}_{self}$ and $\mathcal{N}_{other_1} \ldots \mathcal{N}_{other_K}$
10:         **if** update interval reached **then**
11:             Train `conformalModels` using $\mathcal{N}_{other_j}$'s state-action mappings; for $j \in [1, K]$
12:             Train $N_{self}$ using PPO
13:             Train $N_{other_j}$ using PPO; for $j \in [1, K]$
14:         **end if**
15:     **end for**
16: **end for**
---

Now we formally describe our proposed algorithm — *Conformal Action Modeling-based Multi-Agent Reinforcement Learning* or CAMMARL. Our objective is to inform an $\mathcal{N}_{self}$'s decision-making by modeling the other agent's actions in the environment as conformal prediction sets that contain the true actions with a high probability (for example, 95%). More specifically, $\mathcal{N}_{self}$ uses a separate

---
[2]A tabular version can be found in Section F.1

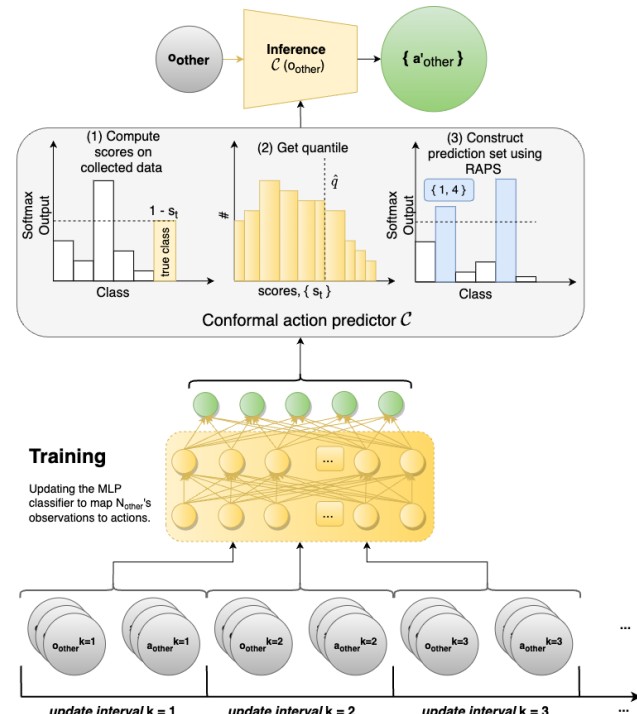

Figure 2: A detailed illustration of conformal action modelling and inference in CAMMARL to generate prediction sets of $\mathcal{N}_{other}$'s actions using conformal predictors.

conformal action prediction model to obtain sets of $\mathcal{N}_{other_j}$'s actions at each timestep that contains latter's true action in the environment at a given time step with high prespecified probabilities.

Algorithm 1 describes the complete workflow of training of agents in CAMMARL. We begin by initializing the actor-critic networks for all the agents in the environment, the conformal model, and the memory buffers for each of these. Now, at the beginning of each episode in the environment, all the agents receive their own partial observations (line 4). Next, the conformal model predicts the actions of all the $\mathcal{N}_{other_j}$'s in the form of a set, which is then provided as an additional input to $\mathcal{N}_{self}$ (lines 6-7), whereas $\mathcal{N}_{other_j}$ has access to only its own partial observation, $o_{other_j}$. The agents now take actions in the environment and continue collecting their experiences (lines 8-9). The agents and the conformal model periodically train using their respective experience memory (lines 10-14).

Figure 2 shows a detailed illustration of our conformal action modeling that materializes internally at each time step. We use only one other agent $\mathcal{N}_{other}$ for simplicity. The conformal predictor $\mathcal{C}$ collects $\mathcal{N}_{other}$'s state-action pairs and periodically learns and updates a neural network classifier, $f(\cdot) : \mathbb{R}^d \to \mathbb{R}^{|\mathcal{A}_{other}|}$ (where $d$ is the number of dimensions in $\mathcal{N}_{other}$'s local observation and $|\mathcal{A}_{other}|$ is the number of possible discrete actions available for $\mathcal{N}_{other}$), to predict action from a given state. Then, we adapt RAPS conformal calibration (Angelopoulos et al., 2020) to our setting. Considering $\mathbf{o} \in O_{other}$ as feature vectors, we use the updated $f$ to compute action probabilities $\hat{\pi}_o \in \mathbb{R}^{|\mathcal{A}_{other}|}$. The probabilities are then ordered from most probable to least probable followed by the estimation of the predictive action set for the given feature inputs. To promote small predictive sets we also add a regularization term as also proposed in RAPS. Formally, for a feature vector $\boldsymbol{o}$ and the corresponding possible prediction $\boldsymbol{a}$, to estimate a set which includes all the actions that will be included before $\boldsymbol{a}$, let us define the total probability mass of the set of actions that are more probable than $a$:

$$\rho_o(a) = \sum_{a'=1}^{|\mathcal{A}_{other}|} \hat{\pi}_o(a') \mathbb{1}_{\{\hat{\pi}_o(a') \geq \hat{\pi}_o(a)\}}$$

Also, if we define a function to rank the possible action outcomes based on their probabilities $\hat{\pi}$ as

$$z_o(a) = |\{a' \in \mathcal{A}_{other} : \{\hat{\pi}_o(a') \geq \hat{\pi}_o(a)\}\}|$$

we can then estimate a predictive action set as follows:

$$\mathcal{C}^*(\mathbf{o}) := \left\{ \mathbf{a} : \rho_{\mathbf{o}}(\mathbf{a}) + \hat{\pi}_{\mathbf{o}}(\mathbf{a}) \cdot u + \lambda \cdot (z_{\mathbf{o}}(\mathbf{a}) - k_{reg})^+ \leq \tau \right\}$$

where $(x)^+$ denotes the positive portion of $x$, and $\lambda, k_{reg} \geq 0$ are regularization hyperparameters to incentivize small set sizes. Here, $u \sim uniform$ [0, 1] (used to allow for randomized procedures) and the tuning parameter $\tau$ (the cumulative sum of the classifier scores after sorting and penalization) to control the size of the sets are identical to as used in RAPS for supervised tasks (Angelopoulos et al., 2020).

To address the data distribution shift across iterations, we continually re-calibrate the conformal model as new data is collected, ensuring that the calibration set remains representative of the current policy and environment dynamics. Thus the model becomes weak as more trajectories are collected until the model is re-calibrated again.

To summarize, in CAMMARL, $\mathcal{N}_{self}$ gets to use estimates of $\mathcal{N}_{other}$'s actions at each time step to make informed decisions in the environment. Instead of modeling exact actions with no uncertainty estimation, we prefer to produce an action set carrying desirable guarantees of containing $N_{other}$'s true action with high probability, integrate it into an agent's downstream task, and enable improved decision-making and collaboration with $\mathcal{N}_{other}$.

## 4 EXPERIMENTS

In this section, we discuss the cooperative tasks with multiple agents used in this study. We note here that though we work in fully cooperative settings in this article, CAMMARL as an idea can be generalized to competitive or mixed settings too (more on this in Appendix).

### 4.1 ENVIRONMENTS

We focus on four cooperative multi-agent environments: **Cooperative Navigation** The agents learn to visit the two landmarks avoiding collisions. In **Level-based Foraging**, agents collect foods based on their levels. In **Pressure Plate**, the agents traverse a grid world and must cooperate by standing on a plate to keep a gate open while the other agents collect the main reward. Finally, we also run CAMMARLin **Google Football**, where 3 agents try to score a goal against a defender and a keeper in a game of football. Further details on these environments are in Section A.

### 4.2 BASELINES

To show the benefits of conformal action set prediction, we compare CAMMARL with the performances of agents in different settings with varying pieces of information made available to $\mathcal{N}_{self}$ during training.

**No-Other-Agent-Modeling (NOAM).** At first, we train $\mathcal{N}_{self}$ without allowing it to model $\mathcal{N}_{other}$. This baseline, as expected, underperforms when compared to any other settings (where any kind of agent modeling is allowed). It is indicative of a lower bound to the learning performance of our model where no kind of benefit from agent modeling is utilized by $\mathcal{N}_{self}$. We call this baseline — *No-Other-Agent-Modeling* or *NOAM*.

**True-Action-Agent-Modeling (TAAM).** Advancing from the inputs available in NOAM, we implement TAAM by allowing $\mathcal{N}_{self}$ to additionally utilize $\mathcal{N}_{other}$'s true actions to train. This baseline helps us evaluate CAMMARL against works that estimate other agents' actions in the environment and use those predictions to enhance the decision-making of their controlled autonomous agents. By giving the true actions as inputs, this baseline can act as an upper bound to such works(He et al., 2016; Grover et al., 2018; Zintgraf et al., 2021; Mealing & Shapiro, 2015; Panella & Gmytrasiewicz, 2017; Albrecht & Ramamoorthy, 2015).

**True-Observation-Agent-Modeling (TOAM).** As discussed in Section 2, learning world models often involves reconstructing observation as an additional task while learning task-related policies (Jain et al., 2022; Hafner et al., 2020; 2021). Inspired by this research, we implement the TOAM baseline where we allow access to $\mathcal{N}_{other}$'s true observations to $\mathcal{N}_{self}$ during training and execution. In other words, we augment $\mathcal{N}_{self}$'s partial observations with the other agent's local observations too. This

baseline can act as an upper bound to the performances of research works that learn to reconstruct states for agents (Hafner et al., 2020; 2021; Wang et al., 2022; Park et al., 2019b; Zhang et al., 2021).

**Global-Information-Agent-Modeling (GIAM).** On the other extreme, we also implement *GIAM*, where $\mathcal{N}_{self}$ trains with complete access to both (1) $\mathcal{N}_{other}$'s true action trajectories ($a_{other}$), and (2) $\mathcal{N}_{other}$'s true observations ($o_{other}$) as additional information. This can be infeasible in real-world scenarios, however, theoretically represents an upper bound on the performance of agents in CAMMARL and other settings.

**Exact-Action-Prediction (EAP).** Building over the inputs of TOAM, we construct a stronger baseline, EAP, in which $\mathcal{N}_{self}$ uses an additional neural network classifier to model a probability distribution over $\mathcal{N}_{other}$'s actions. In other words, instead of predicting conformal sets of actions (like in CAMMARL), in this baseline, $\mathcal{N}_{self}$ tries to model $\mathcal{N}_{other}$'s actions from the latter's observations without accounting for any uncertainty quantification. This baseline is inspired by works that explicitly model the other agent's actions in the environments and utilize them to inform their controlled agent's decision-making (for instance, He et al. (2016); Grover et al. (2018); Zintgraf et al. (2021)). Hence, here, a cross-entropy loss is used to train the added sub-module that predicts the $\mathcal{N}_{other}$'s actions along with a PPO loss to train $\mathcal{N}_{self}$'s policy network.

**CAMMARL.** Now, we implement CAMMARL, where the conformal action prediction model periodically trains on collected observations of $\mathcal{N}_{other}$ and predicts a corresponding conformal set of actions. $\mathcal{N}_{self}$ uses these estimates of $\mathcal{N}_{other}$'s actions along with its own observations and then decides upon its actions in the environment.

### 4.3 RESULTS WITH 2 AGENTS

Figure 3 shows performance curve of all algorithm in environments in Cooperative Navigation and Level Based Foraging. Using additional information, both TAAM and TOAM do reasonably better than NOAM in both tasks. Here, the difference in returns in TOAM and TAAM can be attributed to the fact that the local observations of other agents include more information that can be useful to infer their behavior. For instance, in CN, knowing the relative positions of other agents with respect to the landmarks can be more useful to infer which landmark that agent might be approaching when compared to knowing its current (or history) actions.

GIAM achieves higher returns compared to all other settings in both environments. This is intuitive because it benefits from more information. GIAM is conditioned on $\mathcal{N}_{other}$'s true experiences and consequently demands access to them even during execution. CAMMARL agents are able to distinctly perform better than EAP in LBF, however, interestingly, the performance curve for this baseline nearly overlaps with CAMMARL in CN. Also, in LBF, the curves for TOAM and EAP seem to significantly overlap. We speculate that in a complicated task like LBF, estimating the exact action of $\mathcal{N}_{other}$ can be difficult, and with unaccounted uncertainty in the predictions, $\mathcal{N}_{self}$ suffers from a lower return. In CN, which is comparatively simpler, the closeness of returns in EAP and CAMMARL seem reasonable as even the conformal model predictions eventually start predicting the most probable action with higher probabilities and hence a set of size one (more on this in Section 6).

Figure 3 shows that CAMMARL agents obtain returns that are much closer to the upper bound, *GIAM*, than the lower bound, *NOAM*. Furthermore, CAMMARL's better performance compared to TOAM in both environments can be attributed to the fact that it can be difficult to predict the $\mathcal{N}_{other}$'s intentions by only using $o_{other}$ without any information pertaining to its actions in those situations. And, in TAAM, $\mathcal{N}_{self}$ expected to implicitly encode information regarding $\mathcal{N}_{other}$'s observations from its own local observations or in the latent space and map it to $\mathcal{N}_{other}$'s true actions. We speculate that this could be a strong assumption and consequently very difficult, hence, CAMMARL agents outperform TAAM too. Note here that the sets output by the conformal action prediction model are of varying sizes in each iteration. Now, to be able to use these dynamically changing inputs for $\mathcal{N}_{self}$ in CAMMARL, we convert the output sets to a corresponding binary encoding (by firing up the bits in a zero vector at indices corresponding to the actions predicted by the model). We discuss some more ways to be able to use conformal prediction sets with dynamic sizes and compare CAMMARL's performances in all variations later in the Appendix.

In summary, through experiments in two complex cooperative tasks, we show that (1) CAMMARL indeed works, (2) it outperforms common settings like NOAM, TOAM, and TAAM which assume

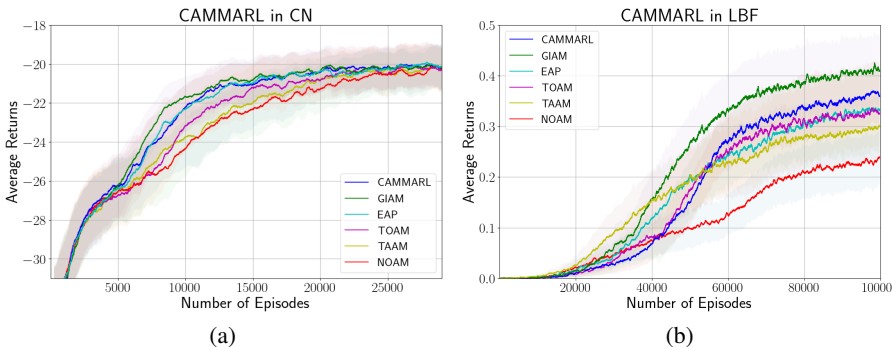

(a)  (b)

Figure 3: Comparison of agent performances (in terms of reward accumulation) in CN (a) and LBF (b) in different settings with varying pieces of information available to $\mathcal{N}_{self}$ during training. CAMMARL's performance is very close to the upper bound, GIAM, and is considerably better than the other extreme, NOAM. It also outperforms the other defined benchmarks (TAAM, TOAM, & EAP) in both tasks, along with the benefit of uncertainty quantification of its estimates. Interestingly, in CN, CAMMARL can be seen to learn arguably faster, but all methods converge to similar results, whereas in LBF, it actually seems to converge to a better policy. The curves are averaged over five independent trials and smoothed using a moving window average (100 points) for readability.

the availability of other agents' true trajectories during training and execution (generally infeasible in real-world scenarios), (3) Its performance is closest to our upper bound of performance (GIAM), (4) CAMMARL agents learn their policies faster than the other baselines, and (5) CAMMARL can be preferred over strong benchmarks such as EAP owing to its higher interpretability due to the theoretical guarantees of conformal predictions in terms of coverage (Angelopoulos et al., 2020) CAMMARL's performance improvements over benchmarks like EAP may be attributed to its use of conformal predictions, which provide well-calibrated uncertainty estimates.

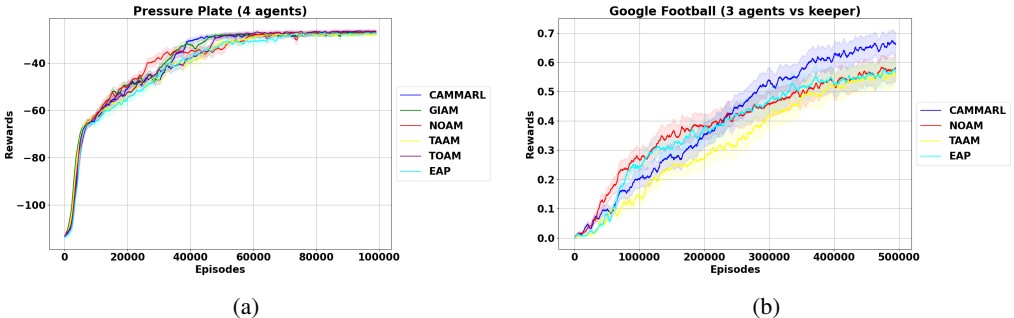

(a)  (b)

Figure 4: Comparison of agent performances (in terms of reward accumulation) in environments with more than 2 agents: (a) Pressure Plate and (b) Google Football. Interestingly, Pressure Plate CAMMARL can be seen to learn arguably faster, but all methods converge to similar results, whereas in Google Football, CAMMARL reaches a higher reward than the baselines. In Google Football, the observations are global, so we did not include TOAM and GIAM. The curves are averaged over five independent runs.

### 4.4 RESULTS WITH MORE THAN 2 AGENTS

CAMMARL can be extended to problems with more than 2 agents, where the main agent uses a conformal model for each of the other agents. From Figure 4 (a) we see that although all the algorithms ultimately converge to the optimal reward, CAMMARL is more sample efficient and converges in roughly 10% fewer episodes compared to GIAM and TOAM. In Google Football (Figure 4 (b)) we notice a considerable improvement as well, with conformal predictions helping in

much better reward accumulation. These experiments also highlight the benefit of using conformal predictions, as TAAM with accurate actions do not converge as fast as CAMMARL.

## 4.5 RESULTS WITH TRAINED AGENTS

| Algorithms | Cooperative Navigation | | | Level Based Foraging | | | Pressure Plate | | |
|---|---|---|---|---|---|---|---|---|---|
| | 50% | 75% | 100% | 50% | 75% | 100% | 50% | 75% | 100% |
| CAMMARL | -20.43 | -20.56 | -20.11 | 0.2 | 0.39 | 0.36 | -30.64 | -30.62 | -30.52 |
| GIAM | -20.64 | -20.47 | -19.87 | 0.32 | 0.33 | 0.42 | -30.47 | -30.58 | -30.76 |
| EAP | -21.23 | -21.32 | -20.44 | 0.22 | 0.25 | 0.3 | -34.24 | -30.21 | -30.18 |
| TOAM | -21.68 | -20.47 | -20.41 | 0.23 | 0.26 | 0.32 | -31.67 | -31.47 | -31.41 |
| TAAM | -22.51 | -20.58 | -21.63 | 0.23 | 0.24 | 0.29 | -36.45 | -32.55 | -32.54 |
| NOAM | -22.87 | -21.48 | -20.47 | 0.09 | 0.19 | 0.24 | -37.69 | -30.89 | -31.94 |

Table 1: Evaluation across 10 episodes during training at 50%, 75%, and 100% of the number of training episodes for the respective environments.

The experimental results highlight CAMMARL's superior convergence rate compared to baseline algorithms across various environments, particularly in Cooperative Navigation and Pressure Plate tasks, where it nearly reaches optimal performance by 50% of training episodes. In Level Based Foraging, CAMMARL outperforms all algorithms at 75% episodes, while other baselines, including GIAM, lag behind. These findings demonstrate CAMMARL's ability to rapidly learn effective strategies in complex multi-agent scenarios, emphasizing its potential for efficient real-world applications.

## 5 MOTIVATING CONFORMAL PREDICTIONS

In Figure 3 (a) and (b), we observe the improved performance of CAMMARL by predicting conformal sets. One key ingredient to CAMMARL was the addition of uncertainty predictions to the actions of the other agents. Thus, we can attribute the increased performance of CAMMARL to this as well. To test this theory, we added one more baseline that is similar to the action prediction baseline (EAP). We call this Action Prediction with Uncertainty (APU), where we add the action prediction probabilities directly into the state space. So both APU and CAMMARL operate on the same information, except APU receives all the predicted actions with their probabilities, but CAMMARL used conformal action sets.

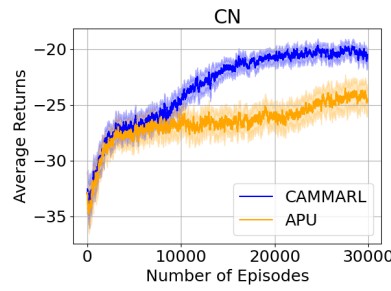

Figure 5: Comparison of agent performances in **CN** with uncertainty information available to $\mathcal{N}_{self}$ during training. This graph highlights the merit of using Conformal Action Predictions over simple uncertainty estimation.

From Figure 5, we can conclude that just adding uncertain predictions is not enough to achieve an uplift in performance that we see for CAMMARL and there is definite merit to using conformal predictions as the way the information is structured makes a difference. Another reason for the poor performance of APU would be that the agent has to parse through the relevant information and learn from it, whereas they are readily provided in a concise manner for CAMMARL.

## 6 DISCUSSION

In this section, we dig deeper and try to analyze the inner components of CAMMARL. Our conformal model constructs prediction sets over a base classification model. Coverage assesses the confidence in the prediction intervals provided by the conformal prediction framework, whereas model accuracy evaluates the predictive capability of the underlying base model used by our conformal model. Through conformal prediction (coverage) we quantify the uncertainty in the predictions from the base model and show that additional knowledge of this information helps agents better model others in

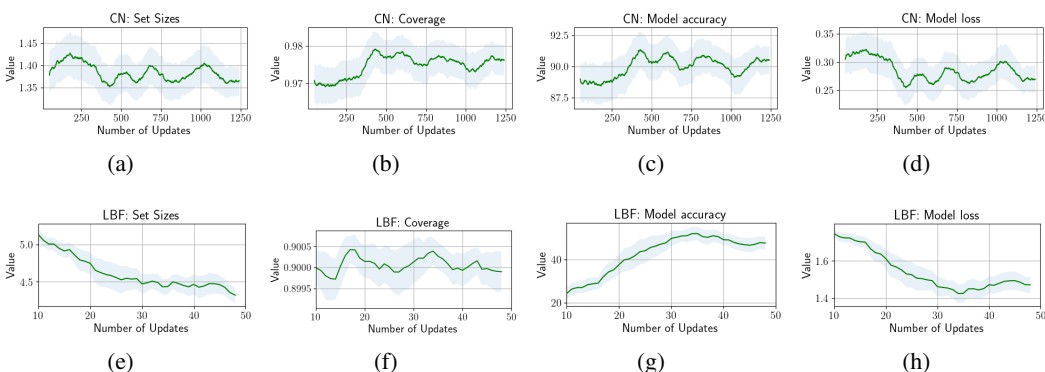

Figure 6: Analysing conformal prediction in CAMMARL over time during the training by looking at trends in conformal sets sizes, coverage of highly probable predictions, model loss and accuracy during training of CAMMARL agents.

the environment. With a base model with low accuracy, to maintain a high, prespecified coverage, our conformal model will end up outputting larger predictive sets. We plot some observable trends during the training of CAMMARL's agents in both the tasks (Figure 6) and discuss each of them here.

**Set Sizes.** We collected the set sizes produced in CAMMARL throughout the training and report them in Figure 6 (a) and 6 (f). Smaller sets are preferred, as they carry specific information which can be more useful practically. The curves show a decreasing trend in the set sizes in CAMMARL in especially in LBF when tracked over the number of updates of the conformal prediction model during training. This is a good sign for CAMMARL, as it shows that the conformal predictions are becoming more precise with continued training over time.

**Coverage.** As also discussed earlier, it is desirable for the predicted sets to provide 1 - $\alpha$ coverage for a pre-defined user-specified $\alpha$ such as 10%. Formally, to map a feature vector, $o_{other} \in \mathcal{O}_{other}$, to a subset of discrete responses, $a'_{other} \in \mathcal{A}_{other}$, it is useful to define an uncertainty set function, $\mathcal{C}(o_{other})$, such that $P(a'_{other} \in C(o_{other})) \geq 1 - \alpha$. Figure 6 (b) shows the increasing trend of confidence coverage in CAMMARL.

**Model accuracy and loss.** In Figure 6 (c) and Figure 6 (d) we show the conformal model's accuracy and loss respectively for CN and LBF in Figure 6 (g) and Figure 6 (h). The model accuracy increases with more data coming in to train over time and the loss correspondingly decreases.

# 7 CONCLUSION

In this article, we propose a novel MARL algorithm, CAMMARL, which calls for confident reasoning about other artificial agents in the environment and benefiting from inferences about their behavior. Through experiments in two cooperative multi-agent tasks, CN and LBF, we showed that guiding an agent's decision-making by inferring other agents' actions in the form of conformal sets, indeed helps in achieving better performances of the learning agents. By using conformal prediction, we were also able to ensure the estimation of predictive sets that covered the real predictions of the intentions of other agents with a very high pre-specified probability of 95%.

**Limitations and Future Works:** In our paper, we analyzed CAMMARL with two agents, however, CAMMARL is certainly generalizable to bigger networks or more simple classifiers, and analyzing its changing performance on varying buffer sizes can help in better comprehending its efficiency. Second, it would be interesting to investigate the CAMMARL's scalability to a system of many agents (say 100 or 1000) or on more complicated multi-agent environments such as tasks requiring a higher need for coordination. Thirdly, our mathematical model in Section 3.2 makes an assumption that the state space is accessible globally which may not be the case in some problems. Finally, in this work, we restricted the agents to infer the behavior of other agents only via conformal sets; it would be interesting to study the cases where more ways of sharing information or modeling agents' behavior are additionally allowed.

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

# APPENDIX

## A  ENVIRONMENTS

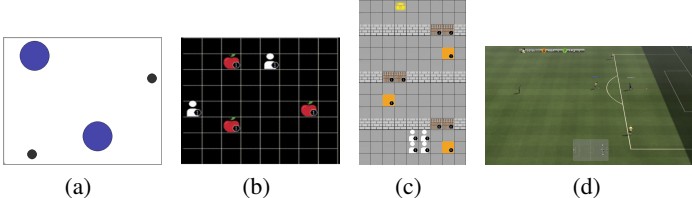

|     |     |     |     |
| --- | --- | --- | --- |
| (a) | (b) | (c) | (d) |

Figure 7: Multi-agent cooperative environments used in this study: (a) OpenAI MPE's **Cooperative Navigation**: Agents (blue) learn to cover the two landmarks (black) avoiding collisions. The figure shows **cooperative navigation** with 2 agents (N=2) and 2 landmarks (L=2) (b) **Level-based foraging**: Agents must collect food and learn to cooperate using sparse rewards. This is a $12 \times 12$ **level-based foraging** grid-world with 2 cooperative players and 4 food locations. (c) **Pressure Plate**: Agents must stand on the pressure plate to keep the gates open for one of the agents to reach the goal. This is a **pressure plate** environment with 4 agents. (d) **Google Football**: 3 players try to score against 1 defender and a goalkeeper in a game of Football.

**Cooperative Navigation (CN) (Mordatch & Abbeel, 2017; Lowe et al., 2017).** In this task, agents are expected to learn to navigate and cover all the landmarks cooperatively and without colliding. Each agent can perceive the other agents and landmarks within its reference frame (in the form of relative positions and velocities) and can take discrete actions to move around (left, right, up, down, stay) in the environment. The agents receive a team reward (so $r_{self}$ and $r_{other}$ are the same in this case) which is calculated as the minimum of the distance of the agents' and landmarks' $(x_i, y_i)$ positions in the grid world. This reward, based on their proximity (or distance) from each of the landmarks, forces the need for cooperation in order to succeed in the task. Furthermore, agents are penalized upon collisions. Formally, the reward function in this environment can be defined as

$$r = \left[ -1 * \sum_{l=1}^{L} min_{\{n \in \mathcal{N}\}}(distance(n, l)) \right] - c$$

where $|\mathcal{N}|$ is the number of agents and L are the landmarks in the environment. Here, c is the number of collisions in an episode and the agents are penalized with -1 for each time two agents collide.

**Level-based foraging (LBF) (Albrecht & Ramamoorthy, 2015).** In this environment, $\mathcal{N}_{self}$ and $\mathcal{N}_{other}$ are part of a $12 \times 12$ grid world which contains four randomly scattered food locations, each assigned a level. The agents also have a level of their own. They attempt to collect food which is successful only if the sum of the levels of the agents involved in loading is greater than or equal to the level of the food. This is a challenging environment, requiring agents to learn to trade off between collecting food for their own and cooperating with the other agent to acquire higher team rewards. Moreover, this environment has sparse rewards making it difficult to learn and operate independently in the environment. In particular, each agent is rewarded equal to the level of the food it managed to collect, divided by its level (its contribution).

**Pressure Plate**[3] In this domain, the agents are expected to learn to co-operate in traversing the gridworld to collect the yellow treasure chest. Each agent is assigned a pressure plate that only they can activate. That agents must stand on the pressure plate to keep the corresponding doorway to open while the other agents can go through it leaving the agent behind. Each agent can only view a limited portion of the environment. If an agent is in the room with their assigned plate, their reward is the negative normalized Manhattan distance between the agent and plate's position, else the reward is the difference between the current and desired room number.

**Google Football Kurach et al. (2020)** For our task, we use the 3 vs 1 with keeper game which is one of the 5 domains in the Football Academy of Google Research Football domain. The main

---
[3]https://github.com/uoe-agents/pressureplate

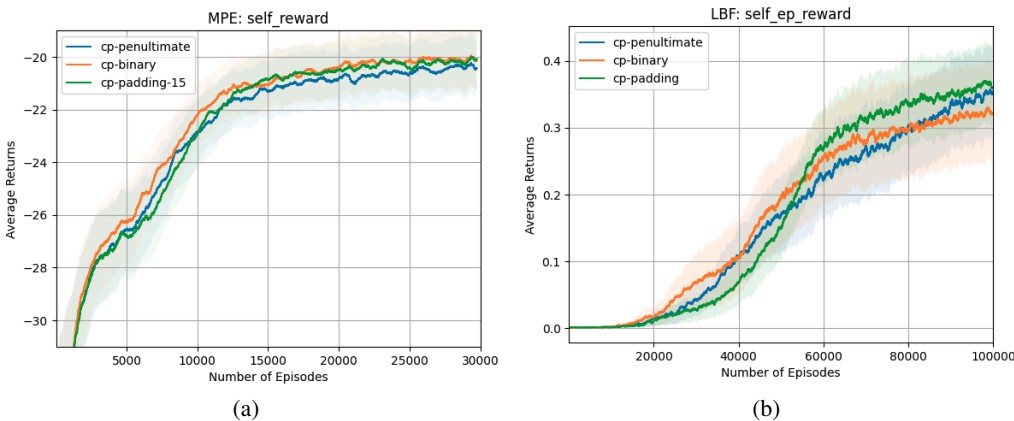

(a)                                              (b)

Figure 8: Comparing the performance of agents when trained with different variants of CAMMARL.

agent controls 3 agents whose objective is to score a goal. The opponent, consisting of 1 defender and a keeper will try to prevent scoring the goal. This environment requires cooperation among the 3 players to score, including skills like passing the ball. The agents receive a reward when it successfully scores a goal past the opponent.

## B    CONFORMAL PREDICTION

Conformal prediction is a statistical framework for constructing prediction intervals or sets with guaranteed coverage probabilities, regardless of the underlying data distribution. This method provides a way to quantify uncertainty in machine learning predictions without making strong distributional assumptions. The key idea behind conformal prediction is the use of a nonconformity measure, which quantifies how different a new example is from previously observed data. This measure is used to determine a prediction set that includes all outcomes sufficiently similar to the observed data. The framework is flexible and can be applied to various machine learning models, including neural networks, support vector machines, and random forests, without modifying their internal workings. One of the main advantages of conformal prediction is its ability to provide valid uncertainty quantification under minimal assumptions. It offers finite-sample guarantees, meaning that the prediction sets are valid not just asymptotically, but also for finite sample sizes. This makes conformal prediction particularly useful in high-stakes applications where reliable uncertainty estimates are crucial for decision-making and autonomous systems.

## C    VARIATIONS OF THE CAMMARL ALGORITHM

As noted earlier, the output of the conformal prediction model, i.e., the conformal sets, are of varying sizes in different situations. So, we came up with numerous ways to be able to use them to implement CAMMARL. In this section, we discuss and compare them and speculate some pros and cons for each of them.

**CAM-PADDING.** The conformal model outputs a set containing the action predictions in the decreasing order of their probability of being the true actions. In other words, the first element in the set is the highest probable action, the next is second highest probable, and so on. The maximum set size being $|\mathcal{A}_{other}|$, we try padding the set with zeros to fix its size when used as input in CAMMARL. We hypothesise that in this way, $\mathcal{N}_{self}$ must be able to learn to infer the information about the actions along with the notion of their importance. Figure 8 also supports this hypothesis by showing the reasonable performance of this version of CAMMARL(green curve). Nevertheless, such padding can be undesirable, particularly because zeros need not necessarily mean "no information", and we try to remove it in the versions discussed next.

**CAM-BINARY.** In this method (also discussed earlier in Section 4), we encode the conformal sets into binary strings. We start with a vector of zeros of size $|\mathcal{A}_{other}|$ and fire up the bits corresponding

to the output actions in the sets. The orange curve in Figure 8 shows that this version of CAMMARL outperforms all the other versions.

**CAM-PENULTIMATE.** Here, we modify CAMMARL to now share the embedding (representations) learned in the conformal prediction model as input to $\mathcal{N}_{self}$. The size of embedding is fixed and this way, padding of conformal action sets in CAMMARL can be avoided. As shown in Figure 8 (blue curve), CAMMARL with representations also manages to obtain returns close to binary-CAMMARL and padded-CAMMARL. We also note here that, binary-CAMMARL and padded-CAMMARL still performs slightly better than CAMMARL with representations.

Interestingly, all versions perform almost equivalently for the use-cases tested in this article. Any method can be used depending on the application at hand. For instance in medical use-cases, the knowledge of ranks of conformal actions is critical, and so cp-padding can be used. In this article, we choose cp-binary. It is a simple way to ensure fixed-sized inputs for $\mathcal{N}_{self}$ in CAMMARL and empirically works very well, and hence we propose this as a generic solution. We note here that, the embeddings here are not being passed in addition to the set of predictions.

## D    CAMMARL IN MIXED SETTINGS

For these experiments, we use the original LBF environment, but we turn off the cooperation flag. This should be a slightly easier task, however, each agent here would fight for different food positions, so it is a mixed setting between cooperation and opposition. Modeling actions of other agents can also help to improve performance in this setting, although it is not as critical as in the cooperative setting.

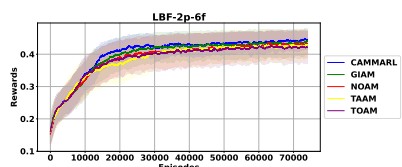

Figure 9 highlights the training performance on LBF with 2 agents respectively. CAMMARL's performance is very close to the upper bound, GIAM, and is better than the other extreme, NOAM. Interestingly, CAMMARL also seems to converge faster than the other baselines. Although this is a simpler task, in which all algorithms roughly converge to the same reward, CAMMARL seems to be on par with our upper bound GIAM, especially in terms of faster convergence.

Figure 9: Comparison of agent performances (in terms of reward accumulation) in LBF with 2 agents (2p) and 6 foods (6f) respectively with cooperation turned off. The curves are averaged over five seeds.

## E    IMPACT OF $\alpha$ ON PERFORMANCE

| $\alpha$ | Reward |
|---|---|
| 0.5 | -21.54 |
| 0.7 | -20.17 |
| 0.9 | -20.11 |
| 0.99 | -21.05 |

Table 2: The table highlights the final evaluation performance of the trained agent acorss various chosen $\alpha$ on in Cooperative Navigation

The ablation study results on Cooperative Navigation with different alpha values provide interesting insights into the performance of CAMMARL.The results suggest an optimal range for alpha exists where CAMMARL performs best. This range balances the trade-off between uncertainty quantification and decision precision: Too much uncertainty (low alpha) may lead to indecisiveness or overly cautious behavior. Too little uncertainty (high alpha) may not provide enough flexibility for agents to adapt to changing situations. The sweet spot (around 0.7-0.9) allows agents to make informed decisions while accounting for a reasonable level of uncertainty in other agents' actions. This ablation study highlights the importance of carefully tuning the conformal prediction parameters in CAMMARL to achieve optimal performance in cooperative multi-agent task.

# F MARL AGENTS: IMPLEMENTATION DETAILS

## F.1 PARTIALLY OBSERVABLE MDP

The partially observable MDP defined and used in this study has numerous parameters as defined previously. Here is a table for quick reference (Table 3).

| Symbol | Description |
|--------|-------------|
| i | *self* or *other* |
| $N_i$ | Agent i |
| S | Set of possible states of the environment |
| $A_i$ | Set of available actions for agent i |
| $O_i$ | Set of local observations of agent i |
| T | State transition function |
| C | Conformal Prediction model |
| $\pi_{\theta_i}$ | Agent i's decision policy |
| $r_i$ | Reward function for agent i |
| $\gamma$ | Discount factor |
| T | Time horizon (length of an episode) |

Table 3: Parameters in our POMDP

We use proximal policy optimization (PPO) Schulman et al. (2017) to update the decision-making policy for both the RL agents, however, any other RL algorithm could be used alternatively. For the individual actor and critic networks, we used 2 fully-connected multi-layer perceptron (MLP) layers. For the conformal prediction model $\mathcal{C}$, we use another fully-connected MLP with 2 layers each with 64 hidden nodes. All the agents and the model $\mathcal{C}$ collect their individual experiences and train their own policies independently.

We add regularization to the conformal model to encourage small set sizes. This regularization is controlled by two hyper-parameters $\lambda$ and $k_{reg}$. Each time the conformal model is trained, we choose the $\lambda$ and $k_{reg}$ parameters from a set that optimizes for small sizes of the action set. The $k_{reg}$ values are dependent on the logits, however, the $\lambda$ values are in a set $[0.001, 0.01, 0.1, 0.2, 0.5]$.

# G  VISUALIZATION OF LEARNED POLICY

In this section, we visualize the learned policy in Pressure Plate with CAMMARL demonstrating cooperation between the 4 agents.

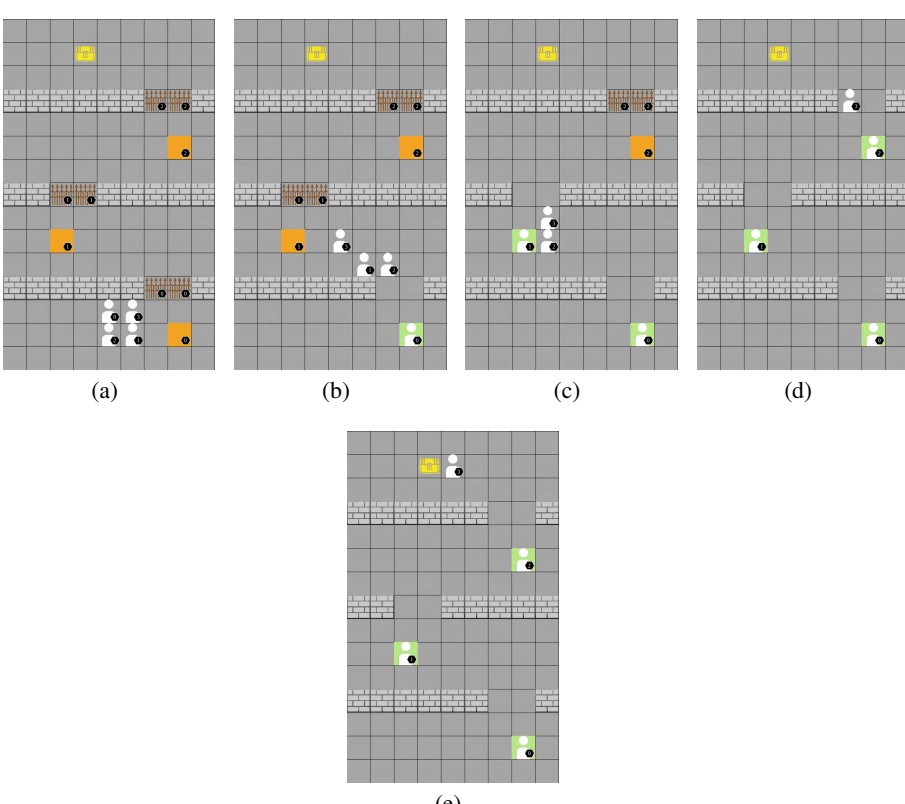

(a)    (b)    (c)    (d)

(e)

Figure 10: Visualization of the policy learned by CAMMARL in Pressure Plate with 4 agents

