# OpenReview forum: "CAMMARL: Conformal Action Modeling in Multi Agent Reinforcement Learning"
_ICLR.cc/2025/Conference — Submitted to ICLR 2025_

### Official Review · Reviewer_W1Jh · 2024-10-26

**Soundness:** 2
**Presentation:** 1
**Contribution:** 2
**Rating:** 3
**Confidence:** 4

**Summary:**

The paper proposes a CAMMARL method, which introduces conformal prediction into opponent modeling to quantify uncertainty in action predictions. The authors conduct experiments on four MARL environments to evaluate the method.

**Strengths:**

1. The paper is well-motivated. The issue of uncertainty prediction in opponent modeling is interesting and noteworthy.
2. This paper introduces conformal prediction to quantify uncertainty in action predictions, which appears technically feasible.

**Weaknesses:**

1. The organization of the paper is poor, especially in the experiments, making it hard to read. I suggest introducing all baselines before discussing the results.
2. The paper lacks comparisons with existing opponent modeling algorithms. Moreover, the algorithm's performance seems to have no obvious advantage in many experimental results.
3. The motivation of the method in Section 3 is not clearly explained. Improving the clarity of the writing would be very helpful.
4. I suggest the authors include some technical background of conformal prediction.

**Questions:**

1. The algorithm demonstrates superior performance on Google Football compared to other tasks. Could you provide some insights into what types of tasks the proposed algorithm may be most effective for?
2. How does the setting of prediction accuracy threshold for selecting the action set impact the algorithm's performance?

---

> ### Author Response · Authors · 2024-11-25
>
> Thank you for your thoughtful review and constructive feedback on our paper. We sincerely appreciate your recognition of our work's motivation and the interesting nature of uncertainty prediction in agent modeling. We have carefully considered your comments and would like to address them as follows:
>
> ### Organization of the paper
> Regarding the paper's organization, we apologize for any confusion caused, especially in the experiments section. We have thoroughly restructured this part, now presenting all baselines before discussing the results, which should significantly improve readability.
>
> ### Comparisons with existing opponent modelling algorithms
> Most of the baselines that perform action modelling will have True Action Agent Modelling as the upper bound, and if we show that our conformal prediction model is better than TAAM, then we are automatically better than the baselines in Line 317. Our primary objective in this study was to introduce and analyze the effectiveness of CAMMARL, particularly in the context of predicting conformal action sets versus single action modeling. Moreover, it's worth noting that CAMMARL can indeed be adapted to integrate with other state-of-the-art MARL techniques that involve action modeling. However, this adaptability wasn't the primary emphasis of our paper.
>
> ### Clarity of motivation in Section 3
> We acknowledge that the motivation in Section 3 could be clearer. We have added a new motivational paragraph at the beginning of this section, explaining why we believe conformal prediction can be a superior alternative for modeling actions of other agents in multi-agent RL environments.
> Background of Conformal Prediction:Thank you for suggesting the inclusion of technical background on conformal prediction. We have added Section B in the appendix, providing a concise description of conformal prediction to aid readers' understanding.
> ### Insights on task suitability
> In cooperative environments, CAMMARL demonstrates particularly impactful performance due to its ability to provide agents with conformal action sets, which enhance decision-making under uncertainty. Tasks such as Google Football and Level Based Foraging require agents to develop strategies collaboratively throughout the episode. In these scenarios, understanding the potential actions of other agents is crucial for effective teamwork. CAMMARL's approach allows agents to model not just single actions but a range of possible actions, incorporating uncertainty into their decision-making processes. This capability enables agents to adapt their strategies dynamically as they gain insights into their teammates' behaviors, ultimately leading to improved coordination and higher overall performance in cooperative tasks.
> ### Impact of prediction accuracy threshold
> While any value of the tuning parameters lambda and k_reg lead to coverage (theoretically proven by (Angelopoulos et al., 2020)), RAPS aims to minimize the set size while maintaining the desired coverage. lambda effectively regularizes the prediction sets by reducing the influence of noisy probability estimates. The conformal confidence level (alpha) is chosen based on the desired coverage. For example, if one aims for 90% coverage, they might adjust the threshold to include sets that achieve this coverage. The choice of alpha influences the size and reliability of the prediction sets; a higher alpha (e.g., 99%) would lead to larger sets that are more likely to contain the true class, while a lower alpha (e.g., 50%) would produce smaller, less reliable sets.
> The ablation study results on Cooperative Navigation with different alpha values provide interesting insights into the performance of CAMMARL:
> Impact of Alpha on Performance:
> |Alpha | Reward |
> | ---- | -------|
> |0.5   | -21.54 |
> |0.7   | -20.17 |
> |0.9   | -20.11 |
> |0.99  | -21.05 |
>
> The results suggest an optimal range for alpha exists where CAMMARL performs best. This range balances the trade-off between uncertainty quantification and decision precision.Too much uncertainty (low alpha) may lead to indecisiveness or overly cautious behavior. Too little uncertainty (high alpha) may not provide enough flexibility for agents to adapt to changing situations. The sweet spot (around 90%) allows agents to make informed decisions while accounting for a reasonable level of uncertainty in other agents' actions.We have added this in Section E of the Appendix.
>
> Given these clarifications and improvements, we kindly request that you consider increasing the score, as we believe these changes address your concerns and strengthen the paper's contribution.

---

> > ### Comment · Reviewer_W1Jh · 2024-11-26
> >
> > Thank you for the authors' response and the additional ablation experiments. After rereading the paper and reading the other reviewers' comments, I believe the current version is not yet ready for acceptance. The paper needs improvements in organization, clarity, and even the quality of the images. Technically, while introducing conformal prediction shows potential, the current evaluations are not convincing enough. Many of the baselines are essentially ablation versions of the proposed algorithm. Including an existing opponent modeling method as a baseline is necessary, which would strengthen the paper’s overall credibility.

---

> > > ### Author Response · Authors · 2024-11-26
> > >
> > > Thank you for your thoughtful feedback and for taking the time to review our paper. We appreciate your insights and would like to address some of the points you raised.
> > >
> > > **Clarification on Organization and Clarity** We understand that you have concerns regarding the organization and clarity of our paper. To better address these issues, could you please provide specific sections or aspects where you found the organization lacking or the clarity insufficient? Your detailed feedback would be invaluable in helping us enhance the readability and coherence of our manuscript. Regarding the quality of figures, could you point out which figure you are referring to? We would be happy to regenerate them.
> > >
> > > **Response to Baseline Concerns** Regarding your comment on the baselines, we would like to clarify that the baselines we included are not merely ablation versions of our proposed algorithm. Instead, they serve as upper bounds for existing agent modeling algorithms. Specifically:
> > > TAAM acts as an upper bound for works such as:
> > >
> > > He et al., 2016: Opponent modeling in deep rein forcement learning. In International conference on machine learning, pp. 1804–1813. PMLR, 2016.
> > >
> > > Grover et al., 2018: Learning policy representations in multiagent systems. In International conference on machine learning, pp. 1802–1811. PMLR, 2018.
> > >
> > > Zintgraf et al., 2021: Deep interactive bayesian reinforcement learning via meta-learning. arXiv preprint arXiv:2101.03864, 2021.
> > >
> > > Mealing & Shapiro, 2015: Opponent modeling by expectation–maximization and sequence prediction in simplified poker. IEEE Transactions on Computational Intelligence and AI in Games, 9(1):11–24, 2015.
> > >
> > > Panella & Gmytrasiewicz, 2017: Interactive pomdps with finite-state models of other agents. Autonomous Agents and Multi-Agent Systems, 31:861–904, 2017.
> > >
> > > Albrecht & Ramamoorthy, 2015: A game-theoretic model and best-response learning method for ad hoc coordination in multiagent systems. arXiv preprint arXiv:1506.01170, 2015.
> > >
> > > TOAM serves as an upper bound on methodologies like:
> > >
> > > Jain et al., 2022:  Learning robust dynamics through variational sparse gating. Advances in Neural Information Processing Systems, 35:1612–1626, 2022.
> > >
> > > These models represent advanced benchmarks that demonstrate the potential of agent modeling techniques in complex environments. We believe that including these baselines provides a comprehensive evaluation of our method's performance relative to other agent modeling algorithms.
> > >
> > > We hope this clarifies our intentions and strengthens the credibility of our evaluations. We are committed to improving our paper based on your feedback and look forward to any further suggestions you might have.
> > > Thank you once again for your constructive comments.

---

> ### Author Response · Authors · 2024-11-30
>
> As the deadline approaches, we wanted to reach out to see if you have any additional concerns regarding our paper. We have made significant revisions based on your feedback and would be happy to clarify any points further. If there are no outstanding issues, we kindly request that you consider increasing your score for our submission.
> Thank you for your time and feedback.

---

> > ### Author Response · Authors · 2024-12-03
> >
> > As the deadline approaches, we wanted to kindly follow up regarding our recent revisions. We would greatly appreciate it if you could take a moment to read through the updates we've made in response to your feedback. If you have any additional concerns, we would be more than happy to address them. However, if everything looks satisfactory, we would be grateful if you could consider increasing your score for our submission. Thank you for your time and your thoughtful review.

---

### Official Review · Reviewer_Nv9z · 2024-11-04

**Soundness:** 3
**Presentation:** 2
**Contribution:** 3
**Rating:** 6
**Confidence:** 3

**Summary:**

This paper introduces CAMMARL (Conformal Action Modeling in Multi-Agent Reinforcement Learning), a novel algorithm for MARL that uses conformal prediction to model the actions of other agents as probabilistic sets, i.e. conformal action sets, to improve cooperative decision-making under uncertainty. Conformal prediction sets that CAMMARL provide are distribution-free confidence intervals for actions and their integration improves adaptability in cooperative multi-agent environments. The proposed framework is evaluated in several multi-agent cooperative tasks where CAMMARL demonstrates improved convergence and performance over baseline approaches.

**Strengths:**

1. **Novel Application of Conformal Predictions for Action Modeling:** CAMMARL´s use of conformal prediction sets provides a novel approach to handle uncertainty in agent actions without distributional assumptions. These conformal prediction sets enable agents to act with quantified confidence which offers an advantage where precise action predictions are difficult. Moreover, conformal action predictions provides better interpretability by offering agents clear confidence intervals on co-agent actions.
2. **Empirical Results:** Initial experiments have supporting results, demonstrating CAMMARL´s faster convergence and higher rewards in cooperative tasks compared to baselines.

**Weaknesses:**

1. **Assumption of Global State Access:** As noted by the authors, CAMMARL assumes full access to the state space, which may limit its applicability in real-world settings. Further exploration of adaptations to handle true partial observability could improve its practical relevance.
2. **Simplistic Experimental Scenarios:** The chosen experimental environments are relatively simple and may not fully capture the complexity of real-world, non-stationary, partially observable environments.
3. **Lack of Theoretical Convergence Analysis:** Although the conformal prediction set approach used in CAMMARL provides interpretability and a bounded measure of control over agent uncertainty through confidence sets, a formal convergence analysis would strengthen the framework´s theoretical foundation.

**Questions:**

1. Given the global state access assumption in the mathematical model, could the authors clarify how CAMMARL would perform in environments with strictly local observations? Are there adaptations or modifications that would allow it to operate effectively under full partial observability?
2. Given the potentially high memory requirements, especially in multi-agent settings with extended time horizons, have the authors analyzed the memory complexity of CAMMARL?

---

> ### Author Response · Authors · 2024-11-25
>
> Thank you for your positive assessment of our work. We appreciate your recognition of our novel approach and will address your concerns as follows:
>
> ### Assumption of global state access
> We thank the reviewer for raising this important point. It is indeed a strong assumption particularly in the case of decentralized multi-agent learning, even with observations the problem is still quite challenging to solve. Also, in problems without ego-centric views, generally, the entire observation is available and the key challenge then is figuring out the actions of the other agents for complete information. We have listed this as a limitation in our Limitations section.
> However, when agents need to communicate with each other, rather than sending the entire state space, if the information can be sent in the form of information-rich conformal action sets, then the performance is actually much better as noted by the better performance of CAMMARL as compared to TOAM which also assumes global state access. So this is not the only factor that improves the performance of CAMMARL agents.
>
> ### Simplistic experimental scenarios
> While our current experiments demonstrate CAMMARL's effectiveness in controlled settings, we agree that exploring more complex environments would be valuable. For example for environment with partial observability, we can integrate CAMMARL with belief state estimation techniques. CAMMARL can be integrated into existing MARL algorithms that focus on environments with non-stationarity and partial observability, this paper does not pursue that objective. Instead, our focus is to underscore the superiority of predicting conformal action sets over single-action modeling which we show by comparison with EAP.
>
> ### Lack of theoretical convergence analysis
> We agree that a formal convergence analysis would strengthen our framework. While a full analysis is beyond the scope of this paper, we have added a brief discussion on the theoretical aspects of CAMMARL's convergence in Line 99.
>
> ### Memory complexity analysis
> CAMMARL is not that memory intensive as opposed to generic action modelling. For Cooperative Navigation, for example, the average time for training each episode for EAP was 10.85 episodes/second while for CAMMARL it was around 11.07 episodes/second. Additionally, the RAM usage for CAMMARL was 595MB while for EAP was 591 MB, so they are pretty similar. In terms of compute resources, it is similar to what any other action modelling algorithm would take.
>
> We believe these clarifications and additions address your concerns and further highlight the strengths and potential of our approach. We kindly request to increase the rating if we have addressed all your concerns.

---

### Official Review · Reviewer_ZpwR · 2024-11-04

**Soundness:** 3
**Presentation:** 2
**Contribution:** 2
**Rating:** 5
**Confidence:** 4

**Summary:**

The paper presents an algorithm CAMMARL, which introduces a conformal prediction model to inform each agent of the confidence sets of actions taken by other agents.
The method allows agents to model other agent’s actions along with a confidence set, leading to more informed policy learning. The authors evaluate CAMMARL in fully cooperative tasks with 2, 3, and 4 agent scenarios. The authors perform a comparison of their method with other baselines with increasing levels of agent modeling or information sharing. The authors also perform experiments in a mixed setting without a cooperation requirement.

**Strengths:**

The paper introduces a novel integration of conformal prediction with multi-agent reinforcement learning (MARL. Providing confidence sets enables agents to make a more meaningful decision when the actions of other agents may not be directly observable.
The paper is well organized and clear in the presentation of the methods used. The prediction module provides an important framework that can be used to create more adaptive and robust multi-agent systems.

**Weaknesses:**

The experimental results could be presented in a more compelling manner. Specifically, the plots in Figures 3a, 4a, and 5 are hard to read due to overlapping lines. Improving the clarity and readability of these figures would make it easier for readers to interpret the findings.

 Additionally, while the paper discusses trends in confidence coverage, this claim could be more robustly supported with quantitative evidence. For example, including success rates and reward values would provide a clearer picture of agent performance improvements during evaluation.

A comprehensive assessment of agent performance with the trained model is missing. Adding metrics such as success rates, average rewards, and perf across varying conditions during post-training evaluation would significantly strengthen the experimental results section.

The claim regarding trends in confidence coverage needs more support, as the current presentation in Figure 7 is inconclusive. Similarly, the accuracy and loss curves in Figure 7 lack clarity.

**Questions:**

Could you clarify how the adapted conformal calibration in CAMMARL differs from existing conformal methods in multi-agent reinforcement learning? Is this calibration uniquely addressing specific challenges or nuances in multi-agent environments?

Improving the visual clarity of the figures and plots would greatly help in interpreting the data. A tabular representation of evaluation metrics would be beneficial.

---

> ### Author Response · Authors · 2024-11-25
>
> Thank you for your constructive feedback. We appreciate your recognition of our novel approach and will address your concerns as follows:
> ### Presentation of experimental results
> We agree that some of the results might be hard to read due to the similar convergence of all the baseline methods. We have added a table with evaluations with trained agents at various points of training to illustrate the utility of our algorithm.
>
> We have included Section 4.4 in the revised paper with all the runs with trained agents across 3 environments. We are currently running Google Football, and we will include the results for the main camera-ready paper once they are done. The main takeaway from the table is how fast CAMMARL converges in comparison to the baseline algorithms. This is particularly evident in Cooperative Navigation and Pressure Plate environments where if we look at the performance at 50% and 75% of the total training episodes, we see that CAMMARL has virtually converged. The same cannot be said for the other baseline algorithms (barring GIAM, which is a clairvoyant algorithm using action and observation information directly). In Level Based Foraging, the difference is even more noticeable with CAMMARL beating all algorithms at 75% of total episodes. This experiment truly highlights the improved performance of CAMMARL.
> ### Clarity regarding Figure 7
> Regarding Figure 7, we have added a paragraph to Section 6 explaining the relationship between coverage, model accuracy, and the conformal prediction framework.
> “Our conformal model constructs prediction sets over a base classification model. Coverage assesses the confidence in the prediction intervals provided by the conformal prediction framework, whereas model accuracy evaluates the predictive capability of the underlying base model used by our conformal model. Through conformal prediction (coverage) we quantify the uncertainty in the predictions from the base model and show that additional knowledge of this information helps agents better model others in the environment. With a base model with low accuracy, to maintain a high, prespecified coverage, our conformal model will end up outputting larger predictive sets.”
> ### Comparison to other conformal methods
> To the best of our knowledge, this is the first paper that introduces conformal methods for action modelling in MARL. Our aim is to suggest that adding a conformal set of actions for modelling agent behaviour can help in learning.
> Our intention was to delve into the nuances of action prediction strategies and demonstrate the advantages of utilizing conformal action sets. Moreover, it's worth noting that CAMMARL can indeed be adapted to integrate with other state-of-the-art MARL techniques that involve action modeling.
>
> ### Evaluation with Trained Agents
> We have also added Section 4.4 with evaluations using trained agents, which highlights the impactful performance of CAMMARL compared to the baselines.
>
> Given these significant improvements and clarifications, we respectfully request that you consider increasing the score. We believe these changes address the concerns and substantially enhance the paper's quality and contribution to the field.

---

> > ### Comment · Reviewer_ZpwR · 2024-11-26
> >
> > Thank you for the authors' response and the updates to the paper. After reviewing the changes, I still find that the paper falls short in addressing the previously mentioned concerns.
> >
> > The addition of Table 1 is appreciated as it compares the performance of CAMMARL with other baselines across 10 episodes at 50%, 75%, and 100% of training. However, the results indicate that CAMMARL performs worse than some baselines at 100% of training, raising questions about the choice and adequacy of the training episode length.
> >
> > Regarding Figure 6, the additional explanation is helpful. However, the sinusoidal patterns in Figures 6a, 6b, 6c, and 6f do not demonstrate a clear increasing or decreasing trend. More update steps are needed to establish a convincing trend and strengthen the analysis.

---

> > > ### Author Response · Authors · 2024-11-26
> > >
> > > Thank you for your detailed feedback and the opportunity to address your concerns. We appreciate your thorough review and would like to clarify some points:
> > >
> > > **Regarding Table 1 and CAMMARL's performance** The only baseline that outperforms CAMMARL at 100% training is GIAM. This is expected as GIAM has access to significantly more information and serves as an upper bound to our algorithm.
> > > In the Pressure Plate scenario, EAP is slightly better but the difference is minimal. We invite you to examine the visualization of the learned policy in Section G of the appendix, which demonstrates that our algorithm has indeed converged.
> > > Compared to TAAM, EAP, and TOAM, which are more relevant as upper bounds to existing agent modeling algorithms in the literature, CAMMARL performs well. Notably, it demonstrates superior sample efficiency, reaching convergence much earlier.
> > >
> > > **Addressing your observations on Figure 6**
> > > We acknowledge your point about the lack of a definitive trend, and we have revised our paper to reflect this more accurately. However, we'd like to highlight:
> > > For Figure 6(a), the set size is already close to 1, which is notably low.
> > > Figure 6(b) and 6(c) shows some increasing trends even though not as huge.
> > > The rapid convergence of our model likely contributes to the absence of more pronounced trends, as these values are already quite favorable. The model coverage in 6(f) is close to 90%, supported by good reward outcomes.
> > > In the other figures, we observe a clear trend towards preferring small set sizes while maintaining increased coverage.
> > >
> > > We believe these points demonstrate the effectiveness of CAMMARL, even with the observed patterns in the figures. We're committed to further clarifying these aspects in our paper and welcome any additional suggestions for improvement.

---

> > > > ### Author Response · Authors · 2024-11-30
> > > >
> > > > As the deadline approaches, we wanted to reach out to see if you have any additional concerns regarding our paper. We have made significant revisions based on your feedback and would be happy to clarify any points further. If there are no outstanding issues, we kindly request that you consider increasing your score for our submission.
> > > > Thank you for your time and feedback.

---

### Official Review · Reviewer_uJEX · 2024-11-05

**Soundness:** 3
**Presentation:** 3
**Contribution:** 3
**Rating:** 8
**Confidence:** 3

**Summary:**

The paper proposes to extend MARL by providing agents with conformal predictions of other agents’ actions. By numerical experiments, the authors demonstrate convincingly that the approach can lead to faster convergence and better policies. They also demonstrate that the conformal prediction outperforms a probabilistic prediction of the actions.

**Strengths:**

The main novel idea is original and seemingly effective. The approach is presented clearly, and the experiments are well-designed. This is a convincing paper.

**Weaknesses:**

I would wish for some more explanation/insights on why conformal predictions would outperform other uncertainty-aware predictions. See questions for more details.

**Questions:**

I would like some more information on why the conformal predictions outperform other approaches based on predictions that include uncertainty. The authors test this in Section 5, but they do not really give enough information on the alternative approach (APU) for me to understand. I have trouble following the arguments in the paragraph starting on line 469, and I would appreciate additional information/clarifications to better follow these arguments.
Related to this: I have difficulties following the sentence in line 423ff: “(5) CAMMARL can be preferred over strong benchmarks such as EAP owing to its higher interpretability due to the theoretical guarantees of conformal predictions in terms of coverage (Angelopoulos et al., 2020) (discussed more in Section 6).” – I am not sure that there is enough evidence for claiming that the theoretical guarantees of conformal predictions are the reason for the better performance. Firstly, these theoretical guarantees are only valid under certain assumptions, and I am not sure they are fulfilled in this case. Secondly, my intuition would be that the reason is that (a) uncertainty information does help and (b) maybe the specific format provided by the conformal predictions is easier to handle than other uncertainty predictions. In any case, the sentence should be weakened and some additional discussion in Section 5 would be appreciated.
In this context, the reference to Section 6 is also unclear, maybe you mean Section 5?

---

> ### Author Response · Authors · 2024-11-25
>
> Thank you for your thoughtful review and positive assessment of our paper. We appreciate your questions and will address them as follows:
> * Clarification on the paragraph starting on line 469:
> We acknowledge that our explanation in the paragraph starting on line 469 could be clearer. We have expanded Section 5 to provide a more detailed explanation of the Action Prediction with Uncertainty (APU) baseline. Additionally, we have highlighted that the way information is structured in CAMMARL significantly contributes to its superior performance, not just the uncertainty information.
> * Regarding the sentence on line 423:
> We agree that the statement is too strong and potentially misleading. We have revised it to clarify our intended meaning: the additional theoretical guarantees of using conformal predictions (which we verified empirically in Figure 7) can potentially have better interpretability as opposed to an action prediction model. We have added a sentence at the end to clear this up:
> "CAMMARL's performance improvements over benchmarks like EAP may be attributed to its use of conformal predictions, which provide well-calibrated uncertainty estimates. While the theoretical guarantees contribute to interpretability, they may not be the sole reason for performance gains."
> * Regarding the intuition behind the good performance of CAMMARL:
> We completely agree with your intuition about CAMMARL's good performance. We have added an explanation in Section 5 to address this point that the way CAMMARL handles it in the form of action sets is the reason for the better performance.

---

### Author Response · Authors · 2024-11-25
**Summary of Improvements and Additional Results**

We sincerely appreciate the constructive feedback provided by the reviewers, which has significantly contributed to enhancing the quality of our paper. Here is a summary of the key changes made in response to their comments:
1. Improved Organization: The experiments section has been restructured to present all baseline algorithms before discussing the results, enhancing overall clarity and readability.
2. Expanded Explanations: We have added detailed explanations in Section 5 regarding the Action Prediction with Uncertainty (APU) baseline and clarified how CAMMARL's structured information contributes to its performance.
3. Clarified Motivation: A new section has been included at the beginning of Section 3 to better articulate the motivation behind using conformal prediction for modeling actions in multi-agent environments.
4. Technical Background Addition: An appendix (Section B) has been added, providing a concise description of conformal prediction to aid reader understanding.
5. Ablation Study Results: We have included a table summarizing the ablation study results on Cooperative Navigation, highlighting the impact of different alpha values on performance.
6. Enhanced Experimental Results Presentation: To improve clarity, we have added a table with evaluations of trained agents at various points during training. This includes results from ongoing experiments in Google Football, which will be included in the final version.
7. Clarification on Conformal Calibration: We expanded our explanation of how CAMMARL's adapted conformal calibration addresses specific challenges in multi-agent environments, particularly focusing on non-stationarity.

These revisions aim to address all concerns raised by the reviewers and strengthen our paper's contribution to the field of multi-agent reinforcement learning. We hope these changes meet your expectations and enhance the overall clarity and impact of our work.

---

### Meta-Review · Area_Chair_Z7if · 2024-12-19

**Metareview:**

This paper proposes CAMMARL, a new algorithm for multi-agent reinforcement learning that uses conformal prediction to model the actions of other agents as probabilistic sets, i.e. conformal action sets, to improve cooperative decision-making under uncertainty. The proposed framework is evaluated in several multi-agent cooperative tasks where CAMMARL demonstrates improved convergence and performance over baseline approaches. The introduction of conformal prediction into MARL, especially for action modeling, is new, and the empirical results are mostly effective. However, there were some major concerns regarding the insufficiency and convincingness of the experimental results when compared with several important baselines, as well as some issues regarding the clarity and presentation of the experimental results. I suggest the authors incorporate the feedback from this round, and prepare for other upcoming machine learning venues.

**Additional Comments On Reviewer Discussion:**

There were some major concerns regarding the insufficiency and convincingness of the experimental results, especially the lack of compelling comparisons with several important baselines. There were also some concerns regarding the clarity and rigor in presentation, as well as the organization of the paper. The authors acknowledged the comments, and provided a few new experiments. However, they were not very satisfying and convincing to fully address the reviewers' concerns.

---

### Decision · Program_Chairs · 2025-01-22

Reject